# Vaccine-Induced Seroreactivity Impacts the Accuracy of HIV Testing Algorithms in Sub-Saharan Africa: An Exploratory Study

**DOI:** 10.3390/vaccines10071062

**Published:** 2022-07-01

**Authors:** Frank Msafiri, Alice Manjate, Sarah Lindroth, Nelson Tembe, Raquel Matavele Chissumba, Victoria Cumbane, Ilesh Jani, Said Aboud, Eligius Lyamuya, Sören Andersson, Charlotta Nilsson

**Affiliations:** 1Department of Microbiology and Immunology, Muhimbili University of Health and Allied Sciences, Dar es Salaam P.O. Box 65001, Tanzania; aboudsaid@yahoo.com (S.A.); eligius_lyamuya@yahoo.com (E.L.); 2Division of Clinical Microbiology, Department of Laboratory Medicine, Karolinska Institutet, 17177 Stockholm, Sweden; charlotta.nilsson@folkhalsomyndigheten.se; 3Faculdade de Medicina, Universidade Eduardo Mondlane, Maputo P.O. Box 257, Mozambique; alimanjate28@gmail.com; 4School of Medical Sciences, Örebro University, 70182 Örebro, Sweden; lindroth.sarah@gmail.com (S.L.); soren.andersson@folkhalsomyndigheten.se (S.A.); 5Instituto Nacional de Saúde, Maputo P.O. Box 3943, Mozambique; nelson.tembenrnt@gmail.com (N.T.); raquelmatavele@gmail.com (R.M.C.); vicumbane17@gmail.com (V.C.); ilesh.jani@gmail.com (I.J.); 6Public Health Agency of Sweden, 17182 Solna, Sweden

**Keywords:** vaccine-induced HIV antibodies, vaccine-induced seroreactivity, HIV misdiagnosis, HIV diagnostic algorithms

## Abstract

The detection of vaccine-induced HIV antibody responses by rapid diagnostic tests (RDTs) may confound the interpretation of HIV testing results. We assessed the impact of vaccine-induced seroreactivity (VISR) on the diagnosis of HIV in sub-Saharan Africa. Samples collected from healthy participants of HIVIS and TaMoVac HIV vaccine trials after the final vaccination were analyzed for VISR using HIV testing algorithms used in Mozambique and Tanzania that employ two sequential RDTs. The samples were also tested for VISR using Enzygnost HIV Integral 4 ELISA and HIV western blot assays. Antibody titers to subtype C gp140 were determined using an in-house enzyme-linked immunosorbent assay (ELISA). The frequency of VISR was 93.4% (128/137) by Enzygnost HIV Integral 4 ELISA, and 66.4% (91/137) by western blot assay (WHO interpretation). The proportion of vaccine recipients that would have been misdiagnosed as HIV-positive in Mozambique was half of that in Tanzania: 26.3% (36/137) and 54.0% (74/137), respectively, *p* < 0.0001. In conclusion, the HIV RDTs and algorithms assessed here will potentially misclassify a large proportion of the HIV vaccine recipients if no other test is used. Increased efforts are needed to develop differential serological or molecular tools for use at the point of care.

## 1. Introduction

Substantial gains have been made in the fight against human immunodeficiency virus (HIV) infection. At the end of 2020, more people living with HIV than ever before were aware of their HIV status (84%), were accessing treatment (73%), and were virally suppressed (66%). Additionally, the number of new HIV infections and AIDS-related deaths worldwide fell by 31% and 47%, respectively, since 2010 [1]. In spite of this remarkable progress, the number of people who acquired HIV in 2020 was three times higher than the global target of having less than 500,000 new HIV infections by the end of 2020. A majority of the new infections occurred in sub-Saharan Africa, where HIV-1 subtype C is predominant [1].

A vaccine is needed to halt HIV transmission [2]. However, after more than 30 years of rigorous research, a safe and effective vaccine against HIV still eludes us [3]. The RV144 regimen remains the only HIV vaccine candidate to demonstrate evidence of protection against HIV. A modest vaccine efficacy of 31.2% was seen three and a half years after vaccination in low-risk volunteers primed with a canarypox vector, ALVAC-HIV (vCP1521) expressing gag, pro, envelope (Env) gp41 and gp120, and boosted with alum-adjuvanted bivalent HIV gp120 [4]. However, this efficacy was not replicated when the RV144 regimen was adapted to subtype C [5]. Additionally, mosaic immunogens that are expressed in adenovirus vectors and boosted with clade C gp140 (the HVTN 705 study) also failed to generate sufficient efficacy to protect high-risk women in sub-Saharan Africa from HIV infection. The HVTN 705 regimen had an efficacy of 25.2% (95% confidence interval; −10.5% to 49.3%) at two years after the first vaccination [6]. Despite the unsatisfactory results, the search for an effective vaccine against HIV continues. New vaccine concepts are being evaluated for efficacy in Europe, America, and Africa [7].

The majority of vaccine recipients in HIV vaccine efficacy trials will develop antibodies to HIV. The detection of these vaccine-induced antibodies in the absence of actual HIV infection is known as vaccine-induced seroreactivity (VISR) [8]. The duration of VISR is unknown, but is greatly influenced by the immunogenicity of the candidate HIV vaccine. In some participants, vaccine-induced HIV antibodies have been detected for more than 23 years after vaccination [9].

A series of HIV vaccine clinical trials were conducted in Sweden, Tanzania, and Mozambique to assess the safety and immunogenicity of a multigene, multi-subtype DNA vaccine candidate (HIV-DNA) boosted with heterologous HIV-1 modified vaccinia virus Ankara-Chiang Mai double recombinant vaccine (HIV-MVA) with or without adjuvanted envelope gp140 protein. The majority of those vaccinated (90–100%) developed detectable antibody responses after completion of the vaccinations [10,11]. Furthermore, anti-Env antibody responses were shown to persist for up to three years [12]. The vaccine recipients frequently reacted in enzyme-linked immunosorbent assays (ELISAs) and western blots used for HIV diagnosis, and infection was therefore ruled out by use of HIV DNA/RNA testing [13].

In order to differentiate between antibodies induced by HIV infection and antibodies induced by an HIV vaccine, it is necessary to know which antigens were included in the vaccine(s) and to have assays set up to allow for this distinction. This is often not the case in routine diagnostic settings. In sub-Saharan Africa, HIV diagnosis relies mainly on the detection of antibodies to HIV using rapid diagnostic tests (RDTs). To improve the accuracy of HIV diagnostic algorithms, two to three RDTs are used in series or in parallel as screening, confirmatory, or tiebreaker tests [14]. The HIV RDTs that are utilized in sub-Saharan Africa include; Determine^TM^ HIV-1/2, SD Bioline HIV1/2, Uni-Gold^TM^ HIV-1/2, ImmunoFlow HIV 1–HIV 2, HIV 1/2 Stat-Pak, Clearview COMPLETE HIV-1/2, Genie II HIV1/HIV-2, First Response HIV 1-2, INSTI HIV-1/HIV-2, and ImmunoComb II HIV 1&2 BiSpot [15,16]. Considering the emotional trauma and social harms caused by HIV misdiagnosis, it is imperative that the RDT-based diagnostic strategies are able to discriminate HIV antibodies elicited by vaccination from those generated in response to an infection [17].

In the present study, given the lack of a licensed differential HIV RDT and the long-term persistence of vaccine-induced HIV antibodies, we explored the impact of VISR on the HIV rapid test algorithms in sub-Saharan Africa and evaluated the diagnostic algorithms used in Mozambique and Tanzania.

## 2. Materials and Methods

### 2.1. Study Design

The HIVIS/TaMoVac consortium conducted a series of phase I/IIa HIV vaccine trials that evaluated the safety and immunogenicity of an HIV-DNA prime HIV-MVA/Env protein boost strategy. The HIV-1 DNA vaccine was composed of seven plasmids encoding gp160 of HIV-1 subtypes A, B, and C, Rev subtype B, p17/p24 gag subtypes A and B, and RTmut subtype B, whereas, the HIV-MVA vaccine was genetically engineered to express gp150 of HIV-1 subtype E and Gag and Pol of subtype A [18]. The vaccinees had received three HIV-DNA priming immunizations followed by two HIV-MVA boosts given with or without envelope gp140 protein. We evaluated the impact of VISR on the accuracy of HIV diagnostic algorithms using serum and plasma samples (*n* = 180) collected from HIVIS/TaMoVac clinical trial vaccinees between 2009 and 2014. The samples were collected four weeks after the final vaccination from participants of HIVIS01/02/05 (*n* = 23) [13], HIVIS07 (*n* = 14) [19], HIVIS03 (*n* = 29) [11], TaMoVac01 (*n* = 14) [20], and TaMoVac02 (*n* = 57) [10] HIV vaccine trials. Additionally, samples collected during the follow up of HIVIS03 vaccinees were used to assess the durability of VISR. Samples collected from vaccinees 16 months (*n* = 23) and three years (*n* = 20) after their final vaccination in the HIVIS03 trial were tested. All of the samples were stored at −80 °C and those tested were selected based on availability. HIV infection in vaccinees was ruled out using HIV RNA PCR.

### 2.2. Evaluation of HIV Rapid Test Algorithms

We assessed the impact of VISR on rapid point-of-care tests used for HIV diagnosis in Mozambique and Tanzania, namely Alere Determine^TM^ HIV-1/2 (Alere Medical Co., Ltd., Tokyo, Japan) and SD Bioline HIV1/2 (Standard Diagnostics Inc., Samcheok Si, Korea), as screening assays, respectively, and Uni-Gold^TM^ HIV-1/2 (Trinity Biotech, Bray, Ireland) as a confirmatory test when the first assay was reactive. Alere Determine^TM^ HIV-1/2 detects antibodies against envelope proteins HIV-1 gp41 and HIV-2 gp36, while SD Bioline HIV1/2 contains three recombinant proteins corresponding to HIV-1 gp41, HIV-1 p24, and HIV-2 gp36. Uni-Gold^TM^ HIV-1/2 detects antibodies to gp41 (HIV-1), gp120 (HIV-1), and gp36 (HIV-2) [21]. All rapid tests were performed according to the manufacturers’ instructions. When the Mozambican HIV testing algorithm was applied, the samples were first screened for VISR using Alere Determine^TM^ HIV-1/2 and reactivity was confirmed using Uni-Gold^TM^ HIV-1/2. When using the Tanzanian algorithm, samples were first tested for VISR using SD Bioline HIV1/2 and reactive samples were retested using Uni-Gold^TM^ HIV-1/2. In both countries, patients were considered HIV-infected if both assays were reactive. The results were read and interpreted independently by two laboratory technologists.

### 2.3. Evaluation of HIV ELISA

The extent to which VISR would affect the performance of HIV ELISA was assessed using Enzygnost^®^ HIV Integral 4 ELISA (Siemens Healthcare Diagnostics Products GmbH, Marburg, Germany). Enzygnost^®^ HIV Integral 4 ELISA detects the HIV-1 p24 antigen and antibodies to HIV-1 gp41 and HIV-2 gp36. HIV testing was performed in accordance with the manufacturer’s instructions.

### 2.4. Assessment of HIV Western Blot

Western blots are used as supplemental assays in HIV diagnosis [22]. The effect of VISR on the performance of HIV western blots was evaluated using the MP Diagnostics™ HIV Blot 2.2 western blot assay (Eschwege, Germany). The criteria established by the American Center for Disease Control and Prevention (CDC), the World Health Organization (WHO), the Consortium for Retrovirus Serology Standardization (CRSS), and the American Red Cross (ARC) were used to interpret the results. Samples were considered VISR-positive when they fulfilled the criteria for positivity, as shown in Table 1 below. Results were VISR-negative when no band other than the serum control was present and were indeterminate in the presence of any other band that did not fulfill the criteria for positive or negative results.

### 2.5. Assessment of Anti-Env Antibody Responses

IgG antibodies binding to subtype C (CN54) gp140 recombinant protein were determined using a standardized in-house ELISA as described previously [13]. Briefly, U96 MaxiSorp Nunc-Immuno ELISA plates (Thermo Scientific, Roskilde, Denmark) were coated with Env proteins at a concentration of 0.5 μg/mL and were incubated overnight at +4–+8 °C. After washing, the plates were blocked with 10% fetal calf serum in phosphate buffered saline. The diluted plasma samples, titrated using two-fold dilutions beginning at 1:100, were then added in duplicates and were incubated overnight at +4–+8 °C. Protein–antibody complexes were detected using rabbit antihuman immunoglobulin G conjugated to horseradish peroxidase, and were visualized by adding *O*-phenylenediamine peroxidase substrate. The reaction was stopped using 2 M H_2_SO_4_, and the optical density was read using dual wavelengths 492 nm and 630 nm. The cutoff was based on each volunteers’ baseline reactivity. The mean of the duplicate optical density values was calculated for both preimmunization and postimmunization samples. A sample was considered positive in a dilution of 1:100 if the absorbance value was more than twice the mean of the preimmunization sample run at a 1:100 dilution. The results were reported as reciprocal values of the highest dilutions giving positive signals.

### 2.6. Statistical Analysis

Data were analyzed using GraphPad PRISM version 9. Fisher’s exact test was used to compare the frequency of VISR in different HIV diagnostic tests and the subsequent proportion of vaccinees who would be misclassified as HIV-infected by the two HIV testing algorithms. The Mann–Whitney U test was used to compare the magnitude of antibody responses to HIV-1 subtype C gp140 between vaccinees with VISR and those without VISR. A *p*-value of <0.05 was considered statistically significant.

## 3. Results

### 3.1. The HIV Diagnostic Algorithms Misclassified a Large Proportion of HIV-Uninfected Vaccine Recipients

The extent by which rapid point-of-care tests, ELISA, and western blot assays would detect vaccine-induced antibodies to HIV in healthy, HIV-uninfected vaccine recipients was evaluated using samples (*n* = 137) collected from vaccinees who had participated in phase I/IIa HIVIS and TaMoVac HIV vaccine trials. Four weeks after the last vaccination, the proportion of HIV-uninfected vaccine recipients that were reactive in SD Bioline HIV-1/2 3.0, Uni-Gold™, and Alere Determine^TM^ HIV-1/2 HIV was 85.4% (117/137), 54.0% (74/137), and 36.5% (50/137), respectively (Figure 1A). When Enzygnost HIV Integral 4 ELISA was used, 93.4% (128/137) of the vaccine recipients were reactive (Figure 1A). Additionally, depending on the interpretation criteria, 22% to 99% of the vaccinees were classified as HIV-positive using the western blot assay (Figure 1B). Applying the HIV diagnostic algorithm used in Mozambique (sequential testing using Determine and Uni-Gold), 26.3% (36/137) of the vaccinees were scored HIV-positive, whereas when applying the algorithm used in Tanzania (sequential testing using Bioline and Uni-Gold), more than half of the vaccinees, 54.0% (74/137), scored HIV-positive and would have received an incorrect HIV diagnosis (Figure 2).

### 3.2. Antigen Reactivity Patterns as Determined by Western Blot

Western blot was performed to define the antigen recognition pattern in the vaccinees’ samples. High antibody reactivity was seen for HIV-1 p24 (100% (137/137), gp160 (99.3% (136/137), and gp120 (66.4% (91/137). Around half of the vaccinees had antibodies to p17 (51.1% (70/137), and low antibody reactivity was observed to pol antigens p31 (19% (26/137) and p51 (5.1% (07/137), as well as to env protein gp41 (3.7% (5/137). Additionally, few vaccinees generated antibody responses to p66 (4.4% (06/137) and p55 (2.2% (03/137). The rate of VISR varied based on the interpretation criteria used: CDC (99.3% (136/137), CRSS (99.3% (136/137), WHO (66.4% (91/137), and ARC (22% (30/137).

### 3.3. Antibody Levels Influenced Misclassification of HIV Status

To further understand the reactivity seen in the HIV diagnostic algorithms, we compared the antibody titers to HIV-1 subtype C envelope between vaccinees exhibiting reactivity and those who were non-reactive. The magnitude of antibody responses to HIV-1 envelope protein gp140 was significantly higher among vaccinees who were misclassified as HIV-positive than in those identified as HIV-negative. Applying the Mozambican HIV diagnostic algorithm, the median anti-Env titer was 12,800 (IQR; 3200–60750) in vaccinees incorrectly diagnosed as HIV-infected compared to 1600 (IQR; 800–3200) in vaccinees that were correctly identified as HIV-uninfected, *p* < 0.0001 (Figure 3A). Similarly, using the Tanzanian algorithm, vaccinees with false-positive HIV results had a median anti-Env titer of 8100 (IQR; 2700–24300) compared to 800 (IQR; 400–1600) in those identified as HIV-negative, *p* < 0.0001 (Figure 3B).

### 3.4. Vaccinees Received Correct HIV Diagnosis Results Three Years after the Last Vaccination

The durability of reactivity in HIV rapid tests was explored using stored samples collected from participants who were vaccinated in the HIVIS03 trial [11] and followed up in the HIVIS06 trial [23]. The samples that were tested for VISR durability were collected at three time points; one month after the second HIV-MVA (*n* = 29), sixteen months after the second HIV-MVA immunization (*n* = 23) and three years after the second HIV-MVA boost (*n* = 20). One month after the final HIV-MVA vaccination, 89.7% (26/29) of the vaccinees were reactive in SD Bioline HIV-1/2 3.0, 41.4% (12/29) in Alere Determine^TM^ HIV-1/2, and 48.3% (14/29) in Uni-Gold^TM^ HIV-1/2 tests. According to the HIV diagnostic algorithms used in Mozambique and Tanzania, 24.1% (7/29) and 48.3% (14/29) of the vaccinees, respectively, would have been classified as HIV-infected (Table 2).

Sixteen months after the last HIV-MVA vaccination, 43.5% (10/23) of the vaccinees were still reactive in SD Bioline HIV-1/2 3.0, 47.8% (11/23) in Alere Determine^TM^ HIV-1/2, and 8.7% (2/23) in Uni-Gold^TM^ HIV-1/2 tests. This meant that fewer vaccinees would have been misclassified as HIV-infected at this time point compared to four weeks after the final vaccination. According to the diagnostic algorithms, only 8.7% (2/23) of the vaccinees would have received false HIV-positive results in either of the algorithms used in Mozambique and Tanzania (Table 2).

Three years after the second HIV-MVA immunization, 30% (6/20) of the vaccinees were reactive in SD Bioline HIV-1/2 3.0, 5% (1/20) in Alere Determine^TM^ HIV-1/2, and none in Uni-Gold^TM^ HIV-1/2 tests. Thus, three years after the receipt of three HIV-DNA and two HIV-MVA vaccinations, none of the vaccinees would have been misclassified as HIV-infected using either of the two HIV diagnostic algorithms (Table 2).

## 4. Discussion

Rapid tests are the primary tools for HIV diagnosis in sub-Saharan Africa [14]. We assessed the frequency of VISR by the RDTs used for HIV diagnosis in Tanzania and Mozambique and applied the two countries’ HIV diagnostic algorithms. Stored serum and plasma samples collected from Tanzanian, Mozambican, and Swedish participants of phase I/IIa HIV vaccine trials were used. The vaccinees had been primed three times with a multigene, multiclade HIV-DNA vaccine and boosted twice with heterologous HIV-MVA vaccine given either alone or together with a subtype C recombinant gp140 protein. We found a high frequency of VISR by the RDTs that would potentially impact HIV diagnosis.

To our knowledge, this is the first study investigating the impact of VISR using RDTs that includes a strategy for confirmatory HIV diagnosis that is used in routine diagnostics in the African region. Earlier VISR reports have come from the monitoring of immunogenicity among candidate HIV vaccines using ELISA, western blots, and line immunoassays. Depending on the vaccine products, delivery technology, and sensitivity of the diagnostic test used, the proportion of vaccinees with detectable antibody responses at the end of HIV vaccine clinical studies has varied extensively, ranging from 0.4% to 100%, with higher VISR rates being seen in immunoassays detecting both HIV antibodies and p24 antigens [24]. Despite raising concerns on the personal and social harms that may be caused by VISR, previous studies have not evaluated the actual performance of HIV diagnostic algorithms in the presence of VISR [8,17,24].

In the present study, at peak immunogenicity (four weeks after the last vaccination), 85.4%, 54%, and 36.5% of vaccinees were reactive in SD Bioline HIV-1/2 3.0, Uni-Gold™ HIV-1/2, and Alere Determine^TM^ HIV-1/2, respectively. The difference in seroreactivity was probably due to the antigens used in the diagnostic assays, reflecting the immunogens used in the vaccine studies. The analysis of antibody responses using western blots showed that all (100%) vaccine recipients generated antibodies to HIV-1 p24 core protein, 99.3% of vaccinees had antibodies to HIV gp160, nearly two-thirds (66.4%) had antibodies to HIV-1 gp120, and very few (3.7%) had antibodies to HIV-1 gp41. All RDTs in this study contain HIV-1 gp41 as an antigen. In addition to the detection of antibodies against HIV-1 gp41, Uni-Gold™ HIV and SD Bioline HIV1/2 3.0 also detect antibodies against HIV-1 gp120 and HIV-1 p24, respectively [21]. Therefore, the high reactivity in SD Bioline HIV-1/2 3.0 is likely to have been driven by the presence of HIV-1 p24 recombinant proteins in the diagnostic assay. Likewise, the incorporation of HIV-1 gp120 antigens in Uni-Gold™ HIV-1/2 would have contributed to the detection of more than half of the participants with vaccine-induced antibodies. Alere Determine^TM^ HIV-1/2, which contains recombinant proteins corresponding to HIV-1 gp41 and HIV-2 gp36, generated a lower VISR rate than other HIV rapid assays.

Vaccine-induced antibodies may confound the accuracy of HIV testing algorithms to correctly identify individuals with true HIV infection [25]. The diagnosis of HIV in Mozambique and Tanzania is based on sequential testing with two different HIV RDTs. As a result of VISR, a significant proportion of vaccinees would have been incorrectly identified as HIV-infected in both countries, with misclassification being twice as frequent when SD Bioline HIV-1/2 3.0 and Uni-Gold™ HIV-1/2 (the Tanzanian HIV testing algorithm) were used for the diagnosis of HIV infection (54%) than when the Alere Determine^TM^ HIV-1/2 and Uni-Gold™ HIV-1/2 (the Mozambican HIV testing algorithm) combination was used (26.3%). This type of diagnostic strategy is common in many countries today, especially in low-income settings. The difference in the accuracy of HIV testing algorithms in preventing the misdiagnosis of vaccine recipients shows the importance of keeping VISR in mind when selecting a diagnostic strategy in areas where HIV vaccine trials have been performed.

The confounding effect of VISR on HIV diagnostics in sub-Saharan Africa is due to the fact that the rapid serological assays detect antibodies [8]. The molecular detection of HIV-RNA can distinguish VISR from true HIV infection. However, the routine use of quantitative PCR for HIV diagnosis in resource-limited settings is not feasible due to a lack of laboratory infrastructure, the complexity of the procedure, and the high cost of the instruments [8,26,27,28]. Since most of the HIV testing in resource-limited settings takes place in clinics with little infrastructure, the availability of simple, low-cost, molecular diagnostic tools that require little or no equipment at the point of care (POC) would avert the misdiagnosis of HIV vaccine recipients. Unfortunately, the available portable detection platforms are semi-automated benchtop instruments that rely on expensive processors and a reagent cold chain, making them incompatible with POC needs in sub-Saharan Africa [27,28].

The failure of the current HIV testing algorithms to discriminate true HIV infection from VISR is a public health concern as thousands of healthy, uninfected volunteers being recruited into phase IIb/III prophylactic HIV vaccine trials in sub-Saharan Africa (and elsewhere) are at risk of being mistakenly identified as HIV-infected after vaccination. A wrong HIV diagnosis can have devastating consequences on individuals and their families. They risk being stigmatized, discriminated against, and criminalized. Some will be abandoned by their spouses, prevented from donating blood, and denied employment, travel opportunities, and health insurance [8,29]. In the era of the “test and treat approach”, the unnecessary initiation of life-long antiretroviral therapy may result in the wastage of scarce resources such as medicines, laboratory reagents for measuring viral load and CD4^+^ T cell counts, and valuable clinic and staff time. Additionally, misdiagnosed individuals may be unnecessarily subjected to the adverse effects of antiretroviral drugs [30]. Unless a rapid point-of-care differential test is developed, false-positive HIV diagnoses will become more prevalent in sub-Saharan Africa once an effective HIV-1 vaccine becomes available and is approved for mass immunization.

In the present study, vaccine recipients with false-positive HIV results had higher levels of vaccine-induced antibody responses to HIV-1 Env than those with HIV-negative test results. The median antibody titers to subtype C gp140 were eight to ten times higher in vaccinees with an incorrect HIV diagnosis. The induction of potent, durable antibody responses to HIV is the main goal for any HIV vaccine candidate [8]. VISR will become more common as combinations of HIV vaccine regimens being evaluated in efficacy trials will elicit stronger and long-lasting antibody responses to HIV.

The durability of VISR is unpredictable, but is influenced by the immunogenicity of the vaccine, the dose administered, the interval between vaccination doses, the number of doses given, and the sensitivity of the diagnostic test [31]. In this study, a three-year durability of VISR was assessed, at which time VISR was still detectable in a few vaccine recipients. However, none of them were considered HIV-infected when HIV diagnostic testing algorithms were applied. The durability of VISR has been reported to increase with the administration of Env-based vaccines [31]. A long-term follow up of healthy volunteers from multiple HIV vaccine trials revealed that VISR persisted for more than 23 years in volunteers who received rgp160 [9].

The present study has limitations. First, a comparatively limited number of samples from recipients of one type of vaccine regimen (the HIV-DNA prime, HIV-MVA with or without the rgp140 boost) was evaluated. Second, the frequency and persistence of VISR are known to be influenced by HIV vaccine constructs. An increased availability of samples from participants vaccinated with diverse vaccine constructs and using different doses and vaccination strategies would have improved our understanding of the impact of HIV gene inserts on the accuracy of HIV RDTs. Third, commercial RDTs contain different HIV antigens, resulting in different VISR rates. The evaluation of more than three HIV RDTs would have increased the amount of evidence regarding the extent to which HIV diagnosis in different countries will be affected by seroconversion after vaccination. However, the RDTs studied here are those most frequently used in the African region. Additionally, prospective field evaluation was not done. Studies have shown that the specificity of some HIV RDTs may vary depending on the type of specimen used for HIV testing [32]. Differences in specificity may alter the frequency of VISR and the subsequent interpretation of HIV testing results. Finally, despite the fact that Env immunogens are associated with the long-term persistence of VISR, we could not assess the longevity of VISR in the vaccinees who received the rgp140 boost. This was due to a lack of post-vaccination follow up requirements in the HIV vaccine trial in which the rgp140 boost was administered.

## 5. Conclusions

In summary, the HIV RDTs and algorithms assessed here will potentially misclassify a large proportion of HIV vaccine recipients if no other test is used. Health care providers in sub-Saharan Africa must be made aware of the confounding effects of VISR on the performance of HIV testing algorithms. Increased efforts are needed to develop simple, affordable, serological or molecular tools that can discriminate VISR from true HIV infection at the point of care.

## Figures and Tables

**Figure 1 vaccines-10-01062-f001:**
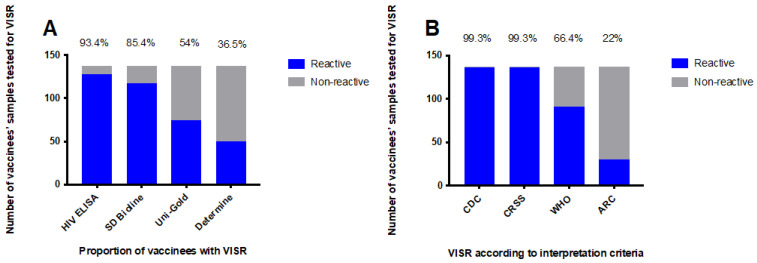
Frequency of VISR. Samples collected four weeks after the final vaccination were evaluated for VISR using rapid point-of-care tests and ELISA (**A**). The highest rates of VISR were seen when using Enzygnost HIV Integral 4 ELISA and SD Bioline HIV-1/2 3.0. HIV misdiagnosis according to western blot interpretation criteria (**B**). The VISR results were interpreted using the criteria detailed in Table 1 (CDC, CRSS, WHO, and ARC criteria). The majority of the vaccinees would have been misclassified as HIV-infected using the CDC and CRSS interpretation criteria.

**Figure 2 vaccines-10-01062-f002:**
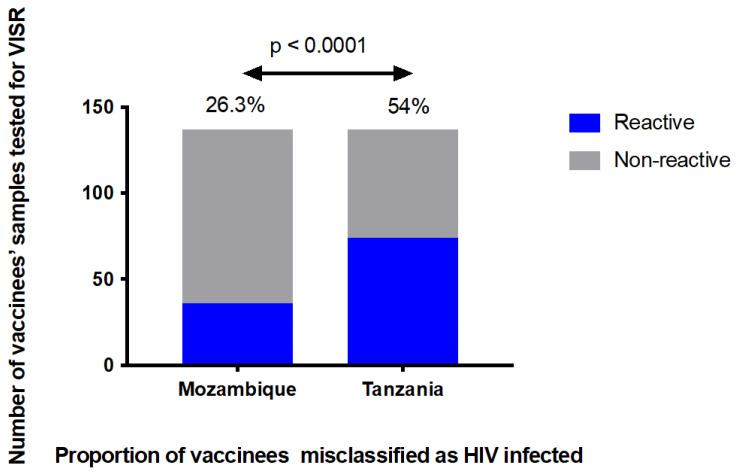
Proportion of vaccinees misdiagnosed as HIV-infected. Samples were tested for VISR using HIV testing algorithms used in Mozambique and Tanzania that use two sequential rapid diagnostic tests: Alere Determine^TM^ HIV-1/2 and Uni-Gold^TM^ HIV-1/2 in Mozambique, SD Bioline HIV1/2 3.0 and Uni-Gold^TM^ HIV-1/2 in Tanzania. Patients were considered HIV-negative if the screening assay was negative and HIV-infected if both assays were reactive. Fisher’s exact test was used to compare the frequency of VISR between the two HIV testing algorithms. The proportion of vaccine recipients who would have been misclassified using the Mozambican HIV testing algorithm was half of that misclassified by the Tanzanian algorithm.

**Figure 3 vaccines-10-01062-f003:**
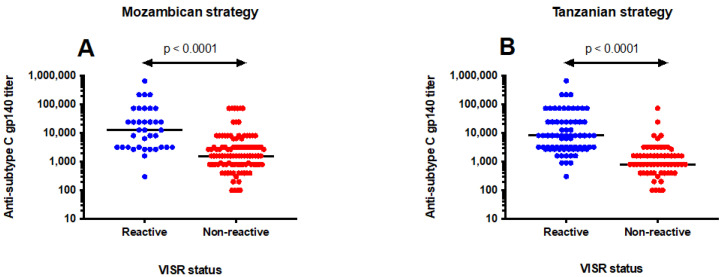
Magnitude of antibody responses among vaccinees. Mann-Whitney U test was used to compare the level of antibody responses between vaccine recipients incorrectly identified as HIV-positive and those who were HIV-negative. In both HIV testing algorithms (**A**,**B**), significantly higher antibody titers were found in vaccinees who were misclassified as HIV-positive than in those classified as HIV-negative. The red and blue circles represent reactive and non-reactive vaccine recipients, respectively.

**Table 1 vaccines-10-01062-t001:** Criteria for positive interpretation of HIV western blot results.

Organization	HIV-Positive
CDC and Association of State and Territorial Public Health Laboratory Directors (ASTPHLD *), 1989 USA	Presence of any two of p24, gp41, gp120/160 bands
WHO, 1990	Presence of two ENV bands with or without GAG or POL
CRSS, 1988 USA	Presence of one ENV band with p24 or p31
ARC, 1988 USA	Presence of one band each of GAG, POL, and ENV

Table adapted from manufacturer’s (MP Biomedicals) product insert. * ASTPHLD was renamed Association of Public Health Laboratories (APHL) in 1998.

**Table 2 vaccines-10-01062-t002:** Frequency of VISR and HIV status misclassifications at one month, sixteen months and three years after the last HIV-MVA immunization.

Diagnostic Assay	Number of Reactive/Number of Tested (%)		
	1 Month after the Second HIV-MVA Vaccination	16 Months after the Second HIV-MVA Vaccination	3 Years after the Second HIV-MVA Vaccination
Alere Determine^TM^ HIV-1/2	12/29 (41.4)	11/23 (47.8)	1/20 (5)
SD Bioline HIV1/2	26/29 (89.7)	10/23 (43.5)	6/20 (30)
Uni-Gold^TM^ HIV-1/2	14/29 (48.3)	2/23 (8.7)	0/20
HIV misdiagnosis using Mozambican algorithm	7/29 (24.1)	2/23 (8.7)	0/20
HIV misdiagnosis using Tanzanian algorithm	14/29 (48.3)	2/23 (8.7)	0/20

In Mozambique, individuals are considered HIV-infected if they are reactive in both Alere Determine^TM^ HIV-1/2 and Uni-Gold^TM^ HIV-1/2, while in Tanzania, they are considered HIV-infected if reactive in both SD Bioline HIV-1/2 3.0 and Uni-Gold^TM^ HIV-1/2.

## Data Availability

Our data came from the testing of stored samples that were collected from the recipients of the HIV-DNA and HIV-MVA/Env vaccines. The study protocol, the statistical analysis plan, de-identified participant data, and the data dictionary will be made available to others upon request. The data will be available from the date of publication and can be shared with investigator support after submission of a data request form, study proposal, a copy of ethical clearance certificate from Muhimbili University of Health and Allied Sciences (https://www.muhas.ac.tz), and a signed data transfer agreement with MUHAS, the primary custodian of this data.

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
