# Peer review of "Vaccine-Induced Seroreactivity Impacts the Accuracy of HIV Testing Algorithms in Sub-Saharan Africa: An Exploratory Study"

_vaccines, 2022, doi:10.3390/vaccines10071062_

Round 1

Reviewer 1 Report

This manuscript by Msafiri et al. describes how HIV vaccine-induced seroconversion (VISR or VISP) can confound the detection of HIV infection following the use of rapid diagnostic testing procedures commonly employed in sub-Saharan Africa. In their study, the authors utilized serum and plasma samples collected at different times post-vaccination from a small pool of recipients of multi-variant Env and Gag DNA vaccines in phase I/IIa clinial trials. These samples were then analyzed using RDTs, ELISAs and CDC/WHO-approved Western blots to either measure vaccine-induced antibody titers or assess the extent to which vaccine recipients could be misdiagnosed as being HIV positive. Overall, the manuscript is very well written, and the scientific and statistical methods employed are appropriately sound. The study is also significant because it brings much needed attention to the emerging concern, initially described in doi: 10.1001/jama.2010.926, of how to accurately diagnose HIV infection in light of increasing enrollment of participants in vaccine trials and a possible near future approval of an effective HIV vaccine. In their Conclusion, the authors propose the development of an alternative RDT or sequential testing approach that can accurately differentiate HIV infections from vaccine-induced antibodies. This will be an arduous R&D endeavor; perhaps the authors could elaborate further on their conclusion by suggesting immunogenic proteins other than Env and Gag or imunogenic viral regions that could be utilized in an alternative RDT to specifically detect HIV infection. The authors may also comment on the feasibility of routinely using RT-PCR, or HIV DNA detection by PCR, as a confirmatory testing approach for vaccine recipients.

Specific comments:

- The legend labeling for the graphs in Figure 1B and 2 are somewhat confusing. It may be more appropriate to use the designation "Reactive or Non-reactive" or "Positive or Negative" to indicate the outcome.

- Can the authors comment on why a statistical comparison was made on the two country outcomes in Figure 2? For instance, are the authors intending to suggest that the sequential testing approach employed in Mozambique would perform better at preventing the misdiagnosis of vaccine recipients?            

Reviewer 2 Report

The manuscript describes the potential misclassification of individuals who have been volunteer-participants in vaccine trials as potentially HIV infected using rapid serological tests. As expected, the vaccinated participants developed antibodies even though the vaccines were not considered efficacious for prevention of infection. The authors have clearly demonstrated that, when they are screened by existing algorithms, these antibodies could result in some individuals being classified or even told that they are HIV+ when in fact they are not infected. While I could see this being a problem under some circumstances, in practice participants usually have evidence – in addition to their memory – of their participation in such trials. An important question is whether these findings have any practical effect: in other words, do the authors know of circumstances, even if anecdotal, where volunteers who were negative (and presumably some were positive and are being treated) were erroneously identified as HIV positive? If there are few or none, then the findings described in this manuscript are mostly theoretical.

In terms of the actual experiments, my only comment is that the number of samples should be clarified: the samples number 180 in the Materials section, yet 137 were tested in section 3.1. Perhaps the remaining samples are those long-term ones later described in section 3.4, but this should be explained.

There were clear differences among the algorithms used in the screening programs. These differences should be highlighted in the discussion, if the authors believe that they are important enough to lead to policy changes.

Reviewer 3 Report

Msafiri et al. have reported on potential interference of antibodies resulting from HIV vaccination trials with serological diagnosis of HIV infections with a particular focus on resource-poor settings, where the diagnosis is based on a combination of serological rapid diagnostic tests. Before publication is considered, I have a number of suggestions to further improve this work.

1. Introduction: As a stylistic element, I suggest shifting the fifth paragraph of the introduction to the position after the third paragraph.

2. Introduction, last sentence: I recommend replacing the term “applied” by “evaluated” to make the intended semantic meaning more clear.

3. Methods chapter: The authors should make it clear whether and in how far the performed assessments were covered by the original ethical clearances of the studies and the provided informed consent statements.

4. Methods chapter: The authors should provide a statement on the age of the samples at the time of testing and on how the samples had been stored. Such information is valuable for the reader to consider pre-analytical effects.

5. Methods chapter, paragraph “Assessment of anti-Env antibody responses”: The authors should state basic test accuracy parameters (at least sensitivity, specificity) of the applied “validated in-house ELISA”. This will make it easier to interpret the obtained results. The sentence on pre- and post-immunization values is difficult to understand. As stated in the “study design” paragraph, “pre-immunization samples” were not available for this study, were they?

6. Statistics: Regarding the chosen significance level, the authors should explain whether or not correction for multiple testing was considered. If it was not considered, please provide a short explanation why.

7. Results/discussion: The authors suggest that measured sero-reactivity might automatically result in an erroneous diagnosis of a HIV-infection. However, as it is totally clear that indirect diagnostic methods like serology allow conclusions on immunologically relevant contacts to structurally related antigens only, this misunderstanding can be easily resolved as long as the tested individual can provide information on his or her participation in a HIV-vaccination trial during the medical history. When such information is available to the diagnosing laboratory, the medical interpretation of a seemingly positive test result would be “non-interpretable” rather than “positive”, wouldn’t it?

8. Discussion, line 304, “test-and-treat” approach) Again, I don’t really understand the authors’ point. If serology is non-reliable due to previous participation in HIV vaccine trials, the logical conclusion to me is that individuals with a respective history of HIV trial participation need molecular HIV testing rather than serological testing to ensure or exclude an actual HIV infection. So, it is unclear to me why the authors do not stress a resulting responsibility of the pharmaceutical enterprises conducting such vaccine trials for ensuring realistic molecular diagnostic options for their trial participants. From the technical point of view, this should be realistically feasible due to broad availability of molecular diagnostic techniques in the post-COVID-era. In the end, it remains a question of money.

9. Conclusions: I disagree with the authors’ conclusion that serological approaches for the discrimination of vaccine-induced and infection-induced antibodies will be the solution, as I fear that such assays will provide increased specificity on the cost of sensitivity. However, excellent sensitivity remains an issue of importance for diagnostic HIV testing as well. Accordingly, molecular diagnostic approaches seem more appropriate to me for such instances. We will need such options anyway. As the authors most likely know, the study leading to the FDA approval of carbotegravir as a long-term PrEP has shown that molecular diagnosis will be necessary to ensure the early diagnosis of HIV infection in carbotegravir PrEP patients. With molecular diagnostic equipment being globally available in the post-COVID-era, it remains hard to see why molecular diagnosis for HIV vaccinees should be basically unfeasible.
